# Contrast-Enhanced Spectral Mammography in the Evaluation of Breast Microcalcifications: Controversies and Diagnostic Management

**DOI:** 10.3390/healthcare11040511

**Published:** 2023-02-09

**Authors:** Luca Nicosia, Anna Carla Bozzini, Giulia Signorelli, Simone Palma, Filippo Pesapane, Samuele Frassoni, Vincenzo Bagnardi, Maria Pizzamiglio, Mariagiorgia Farina, Chiara Trentin, Silvia Penco, Lorenza Meneghetti, Claudia Sangalli, Enrico Cassano

**Affiliations:** 1Breast Imaging Division, Radiology Department, IEO European Institute of Oncology IRCCS, 20141 Milan, Italy; 2Department of Diagnostic Imaging, Radiation Oncology and Hematology, Fondazione Policlinico Universitario “A. Gemelli” IRCCS, 00168 Rome, Italy; 3Department of Statistics and Quantitative Methods, University of Milan-Bicocca, 20126 Milan, Italy; 4Data Management, European Institute of Oncology IRCCS, 20141 Milan, Italy

**Keywords:** CESM, breast cancer, microcalcifications, breast biopsy

## Abstract

The aim of this study was to evaluate the diagnostic performance of contrast-enhanced spectral mammography (CESM) in predicting breast lesion malignancy due to microcalcifications compared to lesions that present with other radiological findings. Three hundred and twenty-one patients with 377 breast lesions that underwent CESM and histological assessment were included. All the lesions were scored using a 4-point qualitative scale according to the degree of contrast enhancement at the CESM examination. The histological results were considered the gold standard. In the first analysis, enhancement degree scores of 2 and 3 were considered predictive of malignity. The sensitivity (SE) and positive predictive value (PPV) were significative lower for patients with lesions with microcalcifications without other radiological findings (SE = 53.3% vs. 82.2%, *p*-value < 0.001 and PPV = 84.2% vs. 95.2%, *p*-value = 0.049, respectively). On the contrary, the specificity (SP) and negative predictive value (NPV) were significative higher among lesions with microcalcifications without other radiological findings (SP = 95.8% vs. 84.2%, *p*-value = 0.026 and NPV = 82.9% vs. 55.2%, *p*-value < 0.001, respectively). In a second analysis, degree scores of 1, 2, and 3 were considered predictive of malignity. The SE (80.0% vs. 96.8%, *p*-value < 0.001) and PPV (70.6% vs. 88.3%, *p*-value: 0.005) were significantly lower among lesions with microcalcifications without other radiological findings, while the SP (85.9% vs. 50.9%, *p*-value < 0.001) was higher. The enhancement of microcalcifications has low sensitivity in predicting malignancy. However, in certain controversial cases, the absence of CESM enhancement due to its high negative predictive value can help to reduce the number of biopsies for benign lesions

## 1. Introduction

Breast cancer (BC) is the most common female cancer, and it is estimated that approximately 1 in 8 women will be diagnosed with BC in their lifetime [1]. As of 2020, BC has the highest incidence of all cancers, with nearly 2.3 million new cases, corresponding to 11.7% of all cancer cases [2]. Furthermore, the incidence of BC has been continuously and slowly increasing over the last ten years [3]. BC accounts for nearly 685,000 deaths among women aged 20 to 59 [2]. However, mortality for this disease has decreased by approximately 42% since 1989, thanks to early diagnosis and improved treatments [4,5]. Screening programs play an essential role in early detection, and the most common and recognized imaging method nowadays is full-field digital mammography (FFDM). Duffy et al. demonstrated a statistically significant reduction in the incidence rate of advanced and fatal BC in women who chose to participate in mammography screening [6]. The American College of Radiology (ACR) recommends that mammography screening for all women should begin as early as at the age of 40, if possible [7,8,9]. FFDM is considered the gold standard for the early detection of breast cancer, even if its performance and sensitivity are impaired in patients with dense breasts with, consequently, higher false-negative rates [10]. Breast density is a meaningful variable that influences the detection of BC in FFDM in screened populations, and there is a four-category classification recommended by the American College of Radiology (ACR) [11]. The loss of sensitivity in patients with dense breasts is often compensated, in clinical practice, by the complementary use of breast ultrasonography, which is useful in identifying those neoplasms that present as nodules [9]. In contrast, the diagnostic performance of mammography, even in dense breasts, is maintained for neoplasms that manifest only as microcalcifications [12]. Calcifications and microcalcifications (<0.5 mm) are very common in breast tissues and are easily detected on mammograms as small bright spots. Mammography is often the only diagnostic method that can identify this type of BC manifestation [13]. Approximately 55% of non-palpable breast cancers occur in the form of microcalcifications without other breast abnormalities, and more than 80% of in situ breast neoplasms occur only as microcalcifications [14]. Microcalcifications are classified as benign or suspicious according to their morphology and distribution [15] and may be the only sign of cancer at a very early stage [12,13,14]. However, their detection, evaluation, and interpretation can sometimes be challenging, even to the most experienced radiologists [16,17,18,19]. Interpreting the microcalcifications seen in mammograms presents a challenge for radiologist [20], who must balance the risk of unnecessary biopsies for benign lesions with the underestimation of potentially pathological microcalcifications [21]. In this context, contrast-enhanced spectral mammography (CESM), a recently developed and increasingly used technique, could provide a valid asset [22]. CESM is a technique derived from mammography (MG) but uses iodinated contrast media and, like contrast-enhanced magnetic resonance imaging (CE-MRI), is based on the principle of cancer angiogenesis. Breast tumors are abundantly vascularized by neoangiogenesis. These vessels allow the contrast media to leak into the tumor interstitium, enhancing the cancer image [23]. CESM’s excellent diagnostic performance in the early identification of breast malignancies has been widely demonstrated in the literature [23,24,25,26,27]. The diagnostic sensitivity appears to be similar to that of breast MRI but with additional advantages, as CESM can be used in patients who cannot undergo MRI due to claustrophobia or MRI-incompatible devices [24]. Furthermore, compared to MRI, CESM requires shorter acquisition times, is better tolerated by patients, is less expensive and more accessible, and appears to be less prone to false positive results [25,26,27].

However, the role of CESM in the interpretation of breast microcalcifications has been scarcely reported in the literature. CESM could be used as a diagnostic method to provide a good ratio of false positives to false negatives in evaluating microcalcifications, thereby reducing the number of biopsies for benign lesions without losing malignant lesions.

The aim of this study was to evaluate the diagnostic performance of the CESM enhancement of recombined images in the prediction of the malignancy of lesions that present as microcalcifications without other radiological findings.

We also wanted to compare the performance of CESM in the prediction of the malignancy of lesions that present as microcalcifications with that of the recombined image enhancement of CESM in the prediction of malignancy of lesions that occur with other types of radiological manifestations such as mass, architectural distortion, and enhancement MRI.

Therefore, one of the study’s primary outcomes was to test whether recombined CESM imaging can help to reduce the number of biopsies required for benign microcalcifications or increase the radiologist’s degree of suspicion of microcalcifications interpreted as benign prior to biopsy.

## 2. Materials and Methods

### 2.1. Study Design and Population

The study was prospective and monocentric. The local Ethics Committees approved this study (Protocol Number IEO S626/311 and IEO 960), and all the patients signed a specific informed consent form.

The enrolment period for this study was from January 2013 to February 2022, and 321 women with 377 suspicious findings were analyzed. All the patients had a standard mammogram and/or an ultrasound, and/or an MRI [28] undertaken for screening in asymptomatic patients.

Following the finding of a suspicious lesion (BIRADS >3), according to the Breast Imaging Reporting and Data System [15], patients were referred for biopsy assessment.

The lesions were classified into microcalcifications without other radiological findings or with other types of radiological manifestations (including mass, mass with microcalcifications, architectural distortion, and enhancement MRI).

In the case of microcalcifications, the patient was referred to the stereotactic-guided procedure. In the case of ultrasound-visible lesions, the patient was referred for an ultrasound-guided biopsy. In case of a lesion visible only on MRI, the patient was referred for MRI-guided biopsy.

Before the biopsy, all the patients enrolled in the study had a CESM.

Briefly, the selection criteria were as follows:a suspicious breast lesion (BIRADS > 3) referred to a cytological or histological examination;a suspicious lesion that had been studied with another conventional diagnostic exam US, MG, or breast MRI;CESM performed prior to cyto/histological assessment.

After CESM acquisition, A.B., an experienced radiologist with more than 25 years of breast imaging experience, qualitatively assessed the enhancement intensity of all the cyto-histologically analyzed breast lesions and gave them a score using the following scale: 0 = no contrast enhancement, 1 = minimal contrast enhancement, 2 = moderate contrast enhancement, and 3 = marked contrast enhancement.

For the first analysis, we considered the unenhanced pattern (score 0) and minimally enhanced pattern (score 1) predictive of benignity and the moderately enhanced pattern (score 2) and markedly enhanced pattern (score 3) predictive of malignancy (Figure 1 and Figure 2).

We also performed a secondary analysis by considering the unenhanced pattern only (score 0) as predictive of benignity and minimally, moderately, and markedly enhanced patterns (scores 1, 2, and 3) as predictive of malignancy.

The B3 lesions were considered benign [29]: with histological assessment, we diagnosed 26 out of 377 B3 lesions. Of the 26 lesions, 7 were surgically removed (for higher radiological suspicion of malignancy, after multidisciplinary discussion), and at subsequent surgery, none of them demonstrated malignancy. The remaining B3 patients were included in the follow-up, and none of them showed malignant neoplasm occurrence over time.

The evaluation of the accuracy in predicting the malignancy of lesions in accordance with the degree of enhancement intensity was carried out by using histological findings as the gold standard. Histologic evaluation of tissue biopsy specimens and on the surgical piece was performed by two pathologists with 15 years of experience in breast histopathology.

The performance of CESM enhancement in the prediction of the malignancy of the lesion was evaluated for both groups (microcalcification without other radiological findings and other types of radiological manifestations). All patients were informed that the outcome of the CESM examination would affect neither the diagnostic-therapeutic procedures nor the prognosis in the case of cancerous lesions. None of the patients included in our study underwent neoadjuvant chemotherapy.

### 2.2. DE-CESM Technique

The CESM examination technique is based on dual-energy breast exposure after the intravenous injection of contrast medium. CESM was performed using Selenia^®^ Dimension^®^ (Hologic Drive, Bedford, MA, USA), the Senographe^®^ Essential full field digital system (GE Healthcare, Chalfont St. Giles, UK), or Amulet^®^ Innovality^®^ (Fujifilm, Akasaka Minato-ku Tokyo, Japan).

First of all, patients were informed by a radiologist about the examination procedure and the associated risks, such as an allergic reaction to iodine-containing contrast medium and exposure to ionizing radiation.

The iodine-based contrast medium was then injected with an automated injection system with an injection flow of 3 mL/s. The type of intravascular iodinated contrast medium administered was Iopromide 370 mg/mL (Ultravist^®^, Manufacturer: Bayer Healthcare Pharmaceutical) at a dose of 1.5 mL/kg and followed by a saline flush. The first image was acquired 2 min after the contrast injection with two different energy exposures (high- and low-energy), and the order of image acquisition was based on the presence of previously detected suspicious findings, with this being: (1) craniocaudal view (CC), suspicious findings-side, (2) mediolateral oblique view (MLO), suspicious findings-side, (3) CC, no suspicious findings-side, and (4) MLO, no suspicious findings-side.

As previously stated, the CESM technique is based on dual-energy exposure: the low-energy images were acquired between 26 and 31 Peak kilovoltage (kVp), while the high-energy images were acquired between 45 and 49 kVp. Two images were then obtained for each view: the low-energy image, a typical mammography image (qualitatively equivalent to full-field digital mammography) [30], and a subtraction image, in which glandular tissues are removed and contrast-enhanced findings can be detected.

A nurse, a radiographer, and a radiologist were present to ensure the safety of the procedure in case of allergic reactions.

### 2.3. Statistical Analysis

Continuous data were reported as median and interquartile ranges (IQR). Categorical data were reported as counts and percentages.

The accuracy in defining the malignant or benign nature of the lesions according to CESM intensity was evaluated using the histological result as the gold standard.

Sensitivity (SE), specificity (SP), positive predictive value (PPV), negative predictive value (NPV), and diagnostic accuracy (DA), with a 95% confidence interval (95% CI), were calculated separated for lesions with microcalcification without other radiological findings and lesions with other types of radiological manifestations.

The SE was calculated as the proportion of lesions correctly evaluated as malignant (according to CESM intensity) out of all lesions which were actually malignant (according to the gold standard, i.e., the histological result). The SP denoted the proportion of lesions correctly evaluated as benign out of all lesions which were actually benign. The PPV reflected the proportion of lesions with a malignant CESM evaluation that were truly malignant according to the histological result. The NPV denoted the proportion of lesions with a benign CESM intensity evaluation that was truly benign according to the histological result. The DA was calculated as the proportion of lesions correctly evaluated as benign or malignant according to CESM intensity.

The chi-square test was used to compare SE, SP, PPV, NPV, and DA among lesions of the two groups.

Considering both the lesions with microcalcifications and with other findings, an ROC curve of the model with the CESM intensity score as an independent variable and the histological result as the dependent variable was performed. The area under the ROC curve (AUC) was also reported.

A *p*-value less than 0.05 was considered statistically significant.

All analyses were performed with the statistical software SAS 9.4 (SAS Institute, Cary, NC, USA).

## 3. Results

Between January 2013 and February 2022, 321 female patients were enrolled in this single-center study. They had a median age at the time of the cyto-histological assessment of 51 years (IQR: 45–59 years). None of them required premedication, experienced allergic reactions to iodinated contrast medium, or required any medical intervention for side effects of the contrast agent. Among the 321 patients, 377 lesions were reported as suspicious (BIRADS > 3), and stereo or ultrasound or MRI-guided cyto-histological assessment was carried out. All these patients underwent CESM before undergoing cyto-histological evaluation. Two hundred and seventy-seven (86.3%) women had only one suspicious lesion, while 44 (13.7%) women had more than one suspicious lesion. Among these women, 28 had lesions in the same breast, while 16 had suspicious lesions in both breasts. A number of the most important descriptive variables of the breast lesion are described in Table 1. In particular, most of the lesions presented as BIRADS 4a (N = 117, 31.4%), in the upper quadrant (226, 59.9%), and were studied with GE mammography (325, 86.2%). Most of the lesions had a dense breast, ACR C (247, 65.5%), and minimal (250, 66.3%) background enhancement.

Of the 377 suspicious lesions, 101 (26.8%) presented as microcalcifications without other radiological findings, while 276 (73.2%) were breast lesions with or without microcalcifications, including mass (N = 249, 66.0%), mass with microcalcifications (N = 10, 2.7%), architectural distortion (N = 8, 2.1%), enhancement MRI (N = 5, 1.3%), and no radiological findings (incidental findings during surgery for other lesions, N = 4, 1.1%). Of these 377 lesions, 109 were not surgically removed, while 268 were surgically removed.

Two hundred forty-nine lesions (66.0%) were histologically malignant, of which 217 (57.6%) were invasive and 32 (8.5%) in situ. The remaining 128 lesions (34.0%) were histologically benign.

Among the 101 lesions that manifested as microcalcifications without other morphological alterations, the pathological results showed that 30 lesions were neoplastic, while 71 were benign. Furthermore, of these 101 lesions, 82 showed a CESM enhancement intensity predictive of benignity (score 0–1), while 19 lesions showed a CESM enhancement intensity predictive of malignancy (score 2–3). The lesion classification according to CESM intensity and histological results among lesions with microcalcification without other radiological findings and lesions with other types of radiological manifestations are reported in Table 2.

The sensitivity (SE) of the degree of contrast enhancement intensity in predicting the malignancy of the lesions with microcalcifications without other radiological findings in CESM was 53.3% (95% CI = 35.5–71.2%); the specificity (SP) was 95.8% (95% CI = 91.1–100%); the positive predictive value (PPV) was 84.2% (95% CI = 67.8–100%); the negative predictive value (NPV) was 82.9% (95% CI = 74.8–91.1%); and the diagnostic accuracy (DA) was 83.2% (95% CI = 75.9–90.5%) (Table 3).

Among the 276 suspicious breast lesions with other types of radiological manifestations, 219 were pathologically malignant, while 57 were benign. Additionally, of these 276 lesions, 87 showed a CESM enhancement intensity predictive of benignity (score 0–1), while 189 lesions showed a CESM enhancement intensity predictive of malignancy (score 2–3) (Table 2).

The sensitivity (SE) of the degree of contrast enhancement intensity in the prediction of malignancy for the CESM of breast lesions with other types of radiological manifestations was 82.2% (95% CI = 77.1–87.3%); the specificity (SP) was 84.2% (95% CI = 74.7- 93.7%); the PPV was 95.2% (95% CI 92.2–98.3%); the NPV was 55.2% (95% CI 44.7–65.6%); and the DA was 82.6% (95% CI = 78.1–87.1%) (Table 3).

Comparing the diagnostic performance of CESM in the analysis of different types of lesion presentations, we found that the sensitivity and the positive predictive value were significantly lower in the evaluation of lesions showing microcalcifications without other radiological findings (*p*-value < 0.001 and 0.049, respectively). In contrast, we found that the specificity and negative predictive value were significantly higher (*p*-value = 0.026 and <0.001, respectively). As a consequence, we found no differences in overall diagnostic accuracy (*p*-value = 0.90) (Table 3).

In the secondary analysis, considering the unenhanced pattern (score 0) as predictive of benignity, and a minimally, moderately, and markedly enhanced pattern (scores 1, 2, and 3, respectively) as predictive of malignancy, among the 101 lesions that manifested as microcalcifications without other morphological alterations, 67 showed a CESM enhancement intensity predictive of benignity, while 34 lesions showed a CESM enhancement intensity predictive of malignancy. Among the 276 lesions with other types of radiological manifestations, 36 showed a CESM enhancement intensity predictive of benignity and 240 of malignancy (Table 4). Specific histological findings are reported in Appendix A.

In this second analysis, the sensitivity (SE) of the degree of contrast enhancement intensity in predicting the malignancy of the lesions with microcalcifications without other radiological findings in CESM was 80.0% (95% CI = 65.7–94.3%); the specificity (SP) was 85.9% (95% CI = 77.8–94.0%); the positive predictive value (PPV) was 70.6% (95% CI = 55.3–85.9%); the negative predictive value (NPV) was 91.0% (95% CI = 84.2–97.9%); and the diagnostic accuracy (DA) was 84.2% (95% CI = 77.0–91.3%) (Table 5).

Comparing the diagnostic performance of CESM in the analysis of different types of lesion presentations, the sensitivity and positive predictive value were significantly lower in the evaluation of lesions showing microcalcifications without other radiological findings (*p*-value < 0.001 and 0.005, respectively). On the contrary, specificity was significantly higher (*p*-value < 0.001). The diagnostic accuracy was similar in the two groups (*p*-value = 0.43).

The ROC curve of the model with the CESM intensity score as the independent variable and the histological result as the dependent variable, considering both the lesions with microcalcifications and with other findings, is reported in Figure 3. The area under the ROC curve (AUC) was 0.902.

## 4. Discussion

CESM has been increasingly tested and used in recent years as it can combine the attributes of the FFDM with one of the essential features on which MRI is based: the enhancement of a lesion [25].

The clinical performance of CESM has been well demonstrated in the literature [25]. In particular, the main applications of this diagnostic method appear to be improving detection rates in dense breasts where mammography loses sensitivity [31,32], increasing the possibility of identifying multicentric neoplasia, and providing a reliable measure of the extent of a neoplasm for appropriate operative planning [33,34].

All the reported studies to date focus on the entire spectrum of breast lesions, not only on one specific subtype, such as suspicious calcifications, without other radiological findings. There are a scattering of studies with small numbers of patients and with controversial results that have evaluated the diagnostic performance of CESM in interpreting those lesions that present solely as microcalcifications [35,36,37]. For example, according to Cheung et al. [36], in a group of 94 patients, CESM provides additional enhancement information with enhancement favorable to the diagnosis of cancer. In contrast, for Houben et al. [35], in a group of 147 patients, CESM is not of added value compared to FFDM in guiding surgical decision-making. This topic is of paramount importance in clinical practice, where a large number of vacuum-assisted biopsies of microcalcifications are often requested for benign lesions as the risk of the misinterpretation of pathologic microcalcificationscannot be allowed [38].

With CESM as the only examination, it is possible to evaluate the microcalcifications from a morphological point of view, using the low-energy image, which is similar to standard mammograms [30], and from a functional point of view, using the subtraction image to evaluate the degree of the enhancement of a lesion. According to what we found in our study, the evaluation of enhancement in patients with lesions that present as only microcalcifications is less sensitive and with a lower PPV than lesions that present in other forms (especially as masses). However, it is more specific without, thereby losing overall diagnostic accuracy. In other words, in this group of patients, we could rely on the intensity of contrast enhancement to detect malignant microcalcifications. For this purpose, we can use low-energy imaging with a conventional BI RADS evaluation [30]. On the other hand, in suspicious cases seen by FFDM or in the low energy image of CESM (e.g., BIRADS 3 and 4a), the lack or low enhancement of microcalcifications, given the high specificity and NPV, may indicate the benignity or the non-invasiveness of the lesions presenting as microcalcifications and avoid unnecessary biopsies. Conversely, in patients with lesions that do not manifest as microcalcifications alone (masses, architectural distortions), a marked or moderated enhancement is more predictive of malignancy due to better sensitivity and PPV.

Thus, the leading clinical utility of the recombined image of contrast medium mammography is associated with its high negative predictive power.

The absence of enhancement in microcalcifications with moderate radiologic risk (BIRADS 3 and 4a) should incline the radiologist to decide in favor of monitoring microcalcifications over time rather than referring them for biopsy.

Following the results of our study, recombined CESM imaging in daily clinical practice could save many biopsies of benign microcalcifications.

The result appears to be superimposable to that of other similar studies performed with breast MRI [39]: however, CESM has the additional advantage of being faster and cheaper and allowing for the morphological evaluation of microcalcifications with low energy imaging in the same examination with a diagnostic power comparable to that of conventional mammography [40].

The limitations of our study are that it was a single-center study with a limited number of patients and the fact that the enhancement features were evaluated by only one radiologist (although one with many years of experience). Further multicenter studies with large numbers of patients are needed to confirm our results.

## 5. Conclusions

The degree of enhancement in lesions presenting as microcalcifications is not sufficiently correlated with the malignancy of the lesions, and for this type of lesion, low-energy imaging can be used with the application of conventional BI-RADS. However, in the most controversial cases, the mild or lack of enhancement of microcalcifications may suggest that they are benign, avoiding unnecessary biopsies. The diagnostic accuracy of CESM enhancement intensity in the prediction of the malignancy of lesions presenting as microcalcifications without other radiological findings compared to lesions presenting with other radiological manifestations is similar. However, the significant difference in sensitivity, positive predictive value, specificity, and negative predictive value between the two groups must be considered in this comparison.

## Figures and Tables

**Figure 1 healthcare-11-00511-f001:**
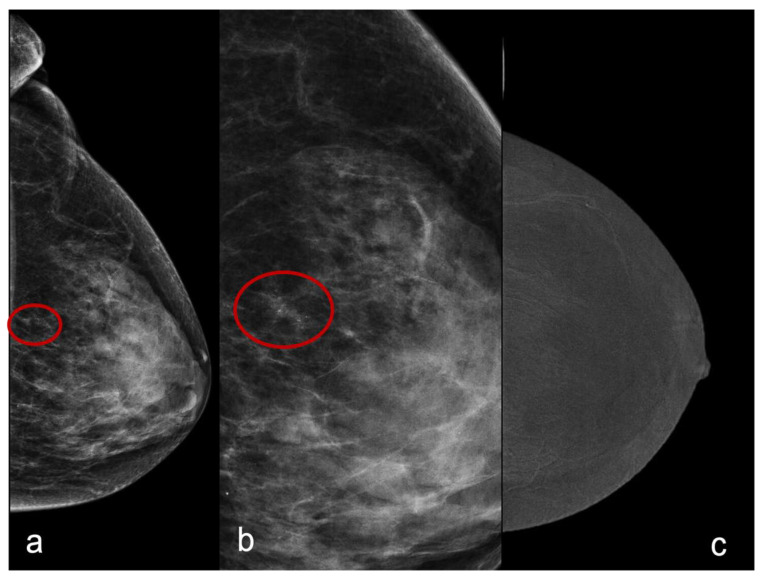
A craniocaudal mammography image of a 56-year-old with a cluster of microcalcifications (**a**). In (**b**), we can appreciate the magnification of the cluster of microcalcifications. There is no evidence of enhancement (score 0) in the CESM subtracted recombined image (**c**). Histology was fibrocystic disease.

**Figure 2 healthcare-11-00511-f002:**
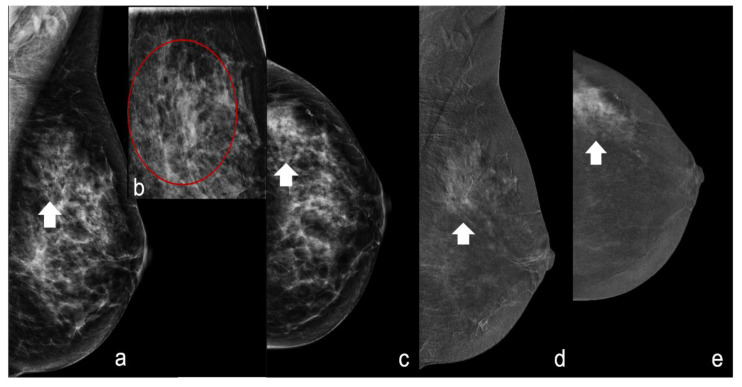
A 45-year-old patient with an extended segmental area of microcalcifications of the upper quadrants of the left breast. (**a**–**c**) Craniocaudal (CC) and mediolateral oblique (MLO) view and magnification of the upper quadrant of the left breast showing suspicious polymorphic microcalcifications (white arrowheads red circle). (**d**,**e**) CESM recombine images (CC and MLO view) showing a marked enhancement (score 3) at the site of suspicious microcalcifications (white arrowheads). Histology was high-grade invasive ductal carcinoma.

**Figure 3 healthcare-11-00511-f003:**
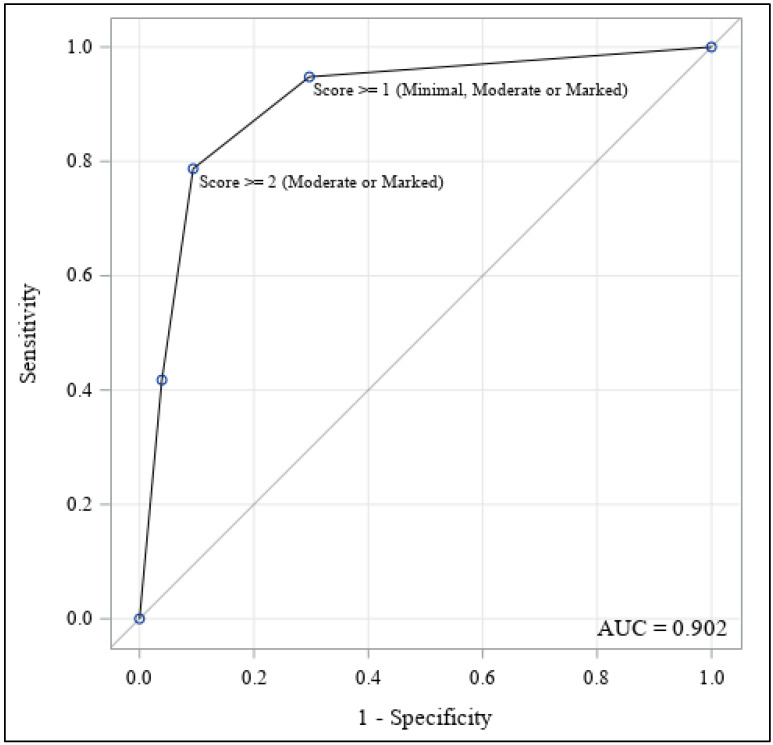
ROC curve (with the AUC) of the model with the CESM enhancement intensity as a predictor of malignancy.

**Table 1 healthcare-11-00511-t001:** Descriptive variables at breast lesion level (N = 377).

Variable	Level	Overall (N = 377)
**Mammograph, N (%)**	Fuji	35 (9.3)
	GE	325 (86.2)
	Hologic	17 (4.5)
**Type of lesion, N (%)**	Microcalcifications	101 (26.8)
	Mass	249 (66.0)
	Mass with microcalcifications	10 (2.7)
	Architectural distortion	8 (2.1)
	Enhancement MRI	5 (1.3)
	Lesion occasionally identified at surgery	4 (1.1)
**Type of lesion, N (%)**	Microcalcifications	101 (26.8)
	No microcalcifications	276 (73.2)
**Quadrant, N (%)**	Lower	68 (18.0)
	Middle	83 (22.0)
	Upper	226 (59.9)
**Side, N (%)**	Left	177 (46.9)
	Right	200 (53.1)
**BIRADS, N (%)**	4a	117 (31.4)
	4b	82 (22.0)
	4c	110 (29.5)
	5	64 (17.2)
	Not applicable (referred to lesion occasionally identified at surgery)	4
**Density (ACR), N (%)**	A	4 (1.1)
	B	79 (21.0)
	C	247 (65.5)
	D	47 (12.5)
**Background, N (%)**	Minimal	250 (66.3)
	Mild	70 (18.6)
	Moderated	35 (9.3)
	Marked	22 (5.8)

**Table 2 healthcare-11-00511-t002:** Lesion classification according to CESM intensity and histological result (benign vs. any malignant) among lesions with microcalcification without other radiological findings (N = 101) and with other types of radiological manifestations (N = 276). The unenhanced and minimally enhanced patterns (scores 0 and 1) were considered predictive of benignity. The moderate and marked enhanced pattern (scores 2 and 3) were considered predictive of malignancy.

	Histological Result					
	with Microcalcifications without Other Radiological Findings			with Other Types of Radiological Manifestations		
Intensity	Benign lesion	Malignant lesion	Total	Benign lesion	Malignant lesion	Total
Benign lesion	68	14	82	48	39	87
Malignant lesion	3	16	19	9	180	189
Total	71	30	101	57	219	276

**Table 3 healthcare-11-00511-t003:** Sensitivity, specificity, positive predictive value, negative predictive value, and diagnostic accuracy among lesions with microcalcification without other radiological findings and with other types of radiological manifestations. Scores 0 and 1 are predictive of benignity, and scores 2 and 3 are predictive of malignancy.

	with Microcalcifications without Other Radiological Findings	with Other Types of Radiological Manifestations	*p*-Value
Sensitivity (SE) [95% CI]	53.3% [35.5–71.2%]	82.2% [77.1–87.3%]	<0.001
Specificity (SP) [95% CI]	95.8% [91.1–100%]	84.2% [74.7–93.7%]	0.026
Positive predictive value (PPV) [95% CI]	84.2% [67.8–100%]	95.2% [92.2–98.3%]	0.049
Negative predictive value (NPV) [95% CI]	82.9% [74.8–91.1%]	55.2% [44.7–65.6%]	<0.001
Diagnostic accuracy (DA) [95% CI]	83.2% [75.9–90.5%]	82.6% [78.1–87.1%]	0.90

**Table 4 healthcare-11-00511-t004:** Lesion classification according to CESM intensity and histological result (benign vs. any malignant) among lesions with microcalcification without other radiological findings (N = 101) and with other types of radiological manifestations (N = 276). The unenhanced pattern (score 0) was considered predictive of benignity. The minimally, moderately, and markedly enhanced patterns (scores 1, 2, and 3, respectively) were considered predictive of malignancy.

	Histological Result					
	with Microcalcifications without Other Radiological Findings			with Other Types of Radiological Manifestations		
Intensity	Benign lesion	Malignant lesion	Total	Benign lesion	Malignant lesion	Total
Benign lesion	61	6	67	29	7	36
Malignant lesion	10	24	34	28	212	240
Total	71	30	101	57	219	276

**Table 5 healthcare-11-00511-t005:** Sensitivity, specificity, positive predictive value, negative predictive value, and diagnostic accuracy among lesions with microcalcifications without other radiological findings and with other types of radiological manifestations. Score 0 is predictive of benignity. Scores 1, 2, and 3 are predictive of malignancy.

	with Microcalcifications without Other Radiological Findings	with Other Types of Radiological Manifestations	*p*-Value
Sensitivity (SE) [95% CI]	80.0% [65.7–94.3%]	96.8% [94.5–99.1%]	<0.001
Specificity (SP) [95% CI]	85.9% [77.8–94.0%]	50.9% [37.9–63.9%]	<0.001
Positive predictive value (PPV) [95% CI]	70.6% [55.3–85.9%]	88.3% [84.3–92.4%]	0.005
Negative predictive value (NPV) [95% CI]	91.0% [84.2–97.9%]	80.6% [67.6–93.5%]	0.13
Diagnostic accuracy (DA) [95% CI]	84.2% [77.0–91.3%]	87.3% [83.4–91.2%]	0.43

## Data Availability

The data presented in this study are available on request from the corresponding author. The data are not publicly available due to privacy concerns, in accordance with GDPR.

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
