# Peer review of "Contrast-Enhanced Spectral Mammography in the Evaluation of Breast Microcalcifications: Controversies and Diagnostic Management"

_healthcare, 2023, doi:10.3390/healthcare11040511_

Round 1

Reviewer 1 Report

Authors have reported qualitative studies, describing it through a radiological perspective. The topic is interesting from the woman's point of view and her health. 

1. I would suggest to the authors improve the discussion section in detail.

2. Try to bring the concrete outcomes of this study

3. Introduction should be updated

4. Improve the quality of figures

5. Can authors explain Table 3 through given parameters and their detailed significance?

Author Response

Authors have reported qualitative studies, describing it through a radiological perspective. The topic is interesting from the woman's point of view and her health.

  1. I would suggest to the authors improve the discussion section in detail.

Thank you for the suggestion. We have edited the discussion trying to better specify the clinical impact of the study.

  1. Try to bring the concrete outcomes of this study
  2. Introduction should be updated

Thanks for the suggestion. We have updated the introduction trying to better focus on the possible clinical impact and outcome of the study

  1. Improve the quality of figures

Thanks for the suggestion.

We submitted the figures in high-resolution TIFF format, 360 dpi, per the journal rules.

  1. Can authors explain Table 3 through given parameters and their detailed significance?

In Table 3 you can find the diagnostic measures (sensitivity, specificity, positive predictive value, negative predictive value, and diagnostic accuracy) for both lesions with microcalcification without other radiological findings and lesions with other types of radiological manifestations.

SE was calculated as the proportion of lesions correctly evaluated as malignant (according to CESM intensity), out of all lesions which are actually malignant (according to the gold standard, i.e., the histological result). SP denoted the proportion of lesions correctly evaluated as benign, out of all lesions which are actually benign. PPV reflected the proportion of lesions with a malignant CESM evaluation which are truly malignant according to the histological result. NPV denoted the proportion of lesions with a benign CESM intensity evaluation which are truly benign according to the histological result. DA was calculated as the proportion of lesions correctly evaluated as benign or malignant according to CESM intensity.

We’ve added the definition of SE, SP, PPV, NPV and DA in the statistical analysis paragraph.

Reviewer 2 Report

The entire study design was carefully designed. The test has no objections. The presented results graphically reflect the consensus of the study.

AI-assisted spectral mammography is increasingly being discussed in scientific publications. In this study, the large number of cases and the selection of patients included in the study speak for the high quality of the study.

Below, in points, I present my questions, comments and suggestions. I ask the authors to refer to each of the issues.

1. In the discussion, I would also refer to other articles on similar techniques using a contrast agent, e.g. "Dynamic contrast-enhanced magnetic resonance imaging (DCE-MRI)". https://www.science.org/stoken/author-tokens/ST-764/full

Although a lot of similar studies have appeared in the last few years, they were carried out on a small number of cases.
I have no objections to the content of the article.

Author Response

The entire study design was carefully designed. The test has no objections. The presented results graphically reflect the consensus of the study.

AI-assisted spectral mammography is increasingly being discussed in scientific publications. In this study, the large number of cases and the selection of patients included in the study speak for the high quality of the study.

Below, in points, I present my questions, comments and suggestions. I ask the authors to refer to each of the issues.

  1. In the discussion, I would also refer to other articles on similar techniques using a contrast agent, e.g. "Dynamic contrast-enhanced magnetic resonance imaging (DCE-MRI)". https://www.science.org/stoken/author-tokens/ST-764/full

Although a lot of similar studies have appeared in the last few years, they were carried out on a small number of cases.

I have no objections to the content of the article.

We thank you for your comments on our article. We also added a brief discussion of the role of breast MRI in evaluating microcalcifications. Studies performed on evaluating breast lesions presenting as pure microcalcifications with CESM are actually few and have controversial results. Therefore, the results of our work can be of excellent clinical utility

Reviewer 3 Report

The authors evaluated the diagnostic performance of CESM in the prediction of malignancy of breast lesions that present as microcalcifications compared with lesions that present with other radiological findings. The enhancement of microcalcifications has low sensitivity in predicting their malignancy. They concluded that, in some controversial cases, the absence of CESM enhancement, due to its high negative predictive value, can help to reduce the number of biopsies for benign lesions. This study is somewhat innovative and has important significance for clinical auxiliary diagnosis and prediction of malignancy of breast lesions. The idea of the trial is clearly thoughtful, the design is reasonable, and the results of the study can support the conclusions. The following changes are required prior to be accepted.

1. This article is mainly a reliability study of methodology. I recommend that the authors combine the results and discussion to be more conducive to reading.

2. Why the authors use a simple 4-point qualitative scale rather than a more detailed quantitative approach. This had a strong impact on the evaluation of this study.

Table 4 should provide detailed patient histological data, such as data on microcalcifications, rather than simple number of people.

4. There are many factors that influence the prediction of malignancy of breast lesions, why not consider some other relevant indicators other than biopsies.

5. The title of the chart needs to be changed, and part of the content can be placed in the table comment.

Author Response

The authors evaluated the diagnostic performance of CESM in the prediction of malignancy of breast lesions that present as microcalcifications compared with lesions that present with other radiological findings. The enhancement of microcalcifications has low sensitivity in predicting their malignancy. They concluded that, in some controversial cases, the absence of CESM enhancement, due to its high negative predictive value, can help to reduce the number of biopsies for benign lesions. This study is somewhat innovative and has important significance for clinical auxiliary diagnosis and prediction of malignancy of breast lesions. The idea of the trial is clearly thoughtful, the design is reasonable, and the results of the study can support the conclusions. The following changes are required prior to be accepted.

1.This article is mainly a reliability study of methodology. I recommend that the authors combine the results and discussion to be more conducive to reading.

Thank you for your comment. We have modified the discussion by dwelling more on the clinical impact of the study based on our results.

2.Why the authors use a simple 4-point qualitative scale rather than a more detailed quantitative approach. This had a strong impact on the evaluation of this study.

Interpretations of the enhancement, as described in the new BIRADS CESM, are also qualitative for ease of interpretation. Therefore, we tried to reproduce the standard clinical practice of CESM interpretation in the study. (Carol et al. CONTRAST ENHANCED MAMMOGRAPHY (CEM) (A supplement to ACR BI-RADS® Mammography 2013). Please refer to page 11: “These are subjective qualitative descriptors…”

Table 4 should provide detailed patient histological data, such as data on microcalcifications, rather than simple number of people.

We thank you for the suggestion. We have added the specific histology results in an additional table.

Supplemental Table 1.

  1. There are many factors that influence the prediction of malignancy of breast lesions, why not consider some other relevant indicators other than biopsies.

We decided to use biopsy as the gold standard result of histological findings because it is the most accurate and objectifiable parameter. Many of the parameters that can be related to a risk of malignancy are shown as descriptive results in Table 1.

  1. The title of the chart needs to be changed, and part of the content can be placed in the table comment.

Thanks. We modified the title of the chart according to your suggestion.

Reviewer 4 Report

In this manuscript, the authors demonstrated that CESM, although a helpful tool to predict malignancy, cannot be used solely. This study will be beneficial to publish on Healthcare after major revisions.

The introduction is very weak and not convincing when describing the significance of the study. Only one sentence was used, “the role of CESM in the interpretation of breast microcalcifications has been scarcely reported in the literature”, but this sentence is very vague.

There is no description on the calculation or analysis process of the sensitivity, specificity, positive predictive value, negative predictive value, and diagnostic accuracy (table 3 and 5), which leads to the weak discussion and conclusions in section 4 and 5. The discussion was not presented in a way that is directly related to the data.

The English grammar and writing of this manuscript need significant improvement. There are too many grammar mistakes and typo. The authors need to be more careful.

Other specific comments on the manuscript are listed below:

1.    The first sentence in the abstract (line 18-19) is very wordy and the grammar needs to be improved. Suggested rephrasing: “…evaluate the diagnostic performance of CESM in predicting breast lesion malignancy due to microcalcifications compared to lesions….”

2.    Line 70, do you mean “evaluation” instead of “valuation”?

3.    Line 71, “very difficult even to the most experienced radiologists.”

4.    Line 72, “presents a challenge for radiologists who must…”

5.    Line 77, “…uses iodinated contrast media and, like contrast-enhanced magnetic…”

6.    Line 81-82, “widely demonstrated in the literature.” Need citation.

7.    Line 91, “compare this performance”. Unclear what “this” means.

8.    Line 100, “from January 2013 to February 2022”.

9.    Line 101, need to be more specific about “suspicious findings”.

10. Line 177, “dose of 1.5 mL/kg and followed by a saline flush”.

11. Line 178, “with two 2 different energy exposures”

12. Line 186, incomplete parenthesis.

Author Response

In this manuscript, the authors demonstrated that CESM, although a helpful tool to predict malignancy, cannot be used solely. This study will be beneficial to publish on Healthcare after major revisions.

The introduction is very weak and not convincing when describing the significance of the study. Only one sentence was used, “the role of CESM in the interpretation of breast microcalcifications has been scarcely reported in the literature”, but this sentence is very vague.

Thanks for the suggestion. We have updated the introduction trying to better focus on the possible clinical impact and outcome of the study.

There is no description on the calculation or analysis process of the sensitivity, specificity, positive predictive value, negative predictive value, and diagnostic accuracy (table 3 and 5), which leads to the weak discussion and conclusions in section 4 and 5. The discussion was not presented in a way that is directly related to the data.

In Table 3 you can find the diagnostic measures (sensitivity, specificity, positive predictive value, negative predictive value, and diagnostic accuracy) for both lesions with microcalcification without other radiological findings and lesions with other types of radiological manifestations.

SE was calculated as the proportion of lesions correctly evaluated as malignant (according to CESM intensity), out of all lesions which are actually malignant (according to the gold standard, i.e., the histological result). SP denoted the proportion of lesions correctly evaluated as benign, out of all lesions which are actually benign. PPV reflected the proportion of lesions with a malignant CESM evaluation which are truly malignant according to the histological result. NPV denoted the proportion of lesions with a benign CESM intensity evaluation which are truly benign according to the histological result. DA was calculated as the proportion of lesions correctly evaluated as benign or malignant according to CESM intensity.

We’ve added the definition of SE, SP, PPV, NPV and DA in the statistical analysis paragraph.

The English grammar and writing of this manuscript need significant improvement. There are too many grammar mistakes and typo. The authors need to be more careful.

Thank you. We have revised the English with a native speaker and apologize for the typo.

Other specific comments on the manuscript are listed below:

  1. The first sentence in the abstract (line 18-19) is very wordy and the grammar needs to be improved. Suggested rephrasing: “…evaluate the diagnostic performance of CESM in predicting breast lesion malignancy due to microcalcifications compared to lesions….”

Thanks, we corrected the text.

  1. Line 70, do you mean “evaluation” instead of “valuation”?

Thanks, we corrected the text.

  1. Line 71, “very difficult even to the most experienced radiologists.”

Thanks, we corrected the text.

  1. Line 72, “presents a challenge for radiologists who must…”

Thanks, we corrected the text.

  1. Line 77, “…uses iodinated contrast media and, like contrast-enhanced magnetic…”

Thanks, we corrected the text.

  1. Line 81-82, “widely demonstrated in the literature.” Need citation.

Thanks, we added the citations.

  1. Line 91, “compare this performance”. Unclear what “this” means.

Thank you we have modified the sentence.

  1. Line 100, “from January 2013 to February 2022”.

Thanks, we corrected the text.

  1. Line 101, need to be more specific about “suspicious findings”.

We highlighted our definition of suspicious finding in the text: Following the finding of a suspicious lesion (BIRADS >3), according to the Breast Imaging Reporting and Data System

  1. Line 177, “dose of 1.5 mL/kg and followed by a saline flush”.

Thanks, we corrected the text.

  1. Line 178, “with two 2 different energy exposures”

Thanks, we corrected the text.

  1. Line 186, incomplete parenthesis.

Thanks we corrected the text.

Round 2

Reviewer 1 Report

Please accept it

Reviewer 4 Report

My concerns were all addressed well in the revision.